# Engaging high-risk groups in early lung cancer diagnosis: a qualitative study of symptom presentation and intervention preferences among the UK's most deprived communities

Grace McCutchan,[1] Julia Hiscock,[2] Kerenza Hood,[3] Peter Murchie,[4] Richard D Neal,[5] Gareth Newton,[6] Sara Thomas,[7] Ann Maria Thomas,[6] Kate Brain[8]

For numbered affiliations see end of article.

**Correspondence to**
Dr Grace McCutchan;
McCutchanGM@cardiff.ac.uk

## ABSTRACT

**Objectives** People at high-risk for lung cancer—current/former smokers, aged 40+ years, with serious lung comorbidity (ie, chronic obstructive pulmonary disease) and living in highly deprived areas—are more likely to delay symptom presentation. This qualitative study aimed to understand the influences on early presentation with lung cancer symptoms in high-risk individuals and intervention preferences.

**Methods** Semi-structured qualitative interviews with 37 high-risk individuals (without a cancer diagnosis), identified through seven GP practices in socioeconomically deprived areas of England, Scotland and Wales (most deprived 20%). A symptom attribution task was used to explore lung symptom perception and help seeking, developed using Leventhal's Common Sense Model. Four focus groups with 16 high-risk individuals and 12 local stakeholders (healthcare professionals and community partners) were conducted to explore preferences for an intervention to promote early lung cancer symptom presentation. Data were synthesised using Framework analysis.

**Results** Individual and area level indicators of deprivation confirmed that interview participants were highly deprived. *Interviews.* Preoccupation with managing 'treatable' short-term conditions (chest infections), led to avoidance of acting on 'inevitable and incurable' long-term conditions (lung cancer). Feeling judged and unworthy of medical help because of their perceived social standing or lifestyle deterred medical help seeking, particularly when difficult life circumstances and traumatic events led to tobacco and alcohol addiction. *Focus groups.* Participants recommended multifaceted interventions in community venues, with information about lung cancer symptoms and the benefits of early diagnosis, led by a trained and non-judgemental facilitator.

**Conclusions** This study was novel in engaging a high-risk population to gain an in-depth understanding of the broader contextual influences on lung cancer symptom presentation. Perceived lack of health service entitlement and complex lives facilitated avoidance of recognising and presenting with lung cancer symptoms. Community-based interventions have the potential to empower disadvantaged populations to seek medical help for lung symptoms.

### Strengths and limitations of this study

► This was the first study to use in-depth qualitative methods to explore how to engage high-risk individuals from socioeconomically deprived areas in early lung cancer diagnosis.

► A major strength of this study was the proactive and rigorous sampling procedures used to ensure that our sample was at high risk for lung cancer.

► Assessment of individual and area level indicators of deprivation confirmed that interview participants were highly deprived; all participants resided in the 20% most deprived areas of the three UK nations, and most participants were unemployed/seeking benefits and/or rented social housing.

► To overcome the methodological limitations associated with studying anticipated or retrospectively recalled cancer symptom presentation, we recruited participants with no previous diagnosis of lung cancer and did not mention lung cancer in the interview study materials or during completion of the symptom attribution task.

► Opportunistic recruitment of focus group participants who may have been more favourably disposed to an intervention was a potential study limitation.

## INTRODUCTION

Lung cancer is the leading cause of cancer mortality worldwide.[1] Outcomes are among the poorest for all cancers, with only 13% of patients with lung cancer surviving five or more years in the UK.[2] Diagnosis of lung cancer at an early stage can enable curative surgical resection, meaning over 80% of patients will survive one year or more when diagnosed at stage I.[3] Delayed medical help seeking for symptoms and the high proportion of lung

**BMJ**

cancer diagnoses through emergency departments may partly explain why lung cancer is commonly diagnosed at an advanced, incurable stage.[4] Due to low specificity of lung cancer symptoms and similarity to other acute and comorbid respiratory conditions, patients face difficulty in knowing when to seek medical help.[5 6]

Multiple symptoms and risk factors for lung cancer including older age, smoking, the presence of a lung comorbidity and socioeconomic deprivation increase the likelihood that a patient presenting to their GP with symptoms indicative of lung cancer will receive a cancer diagnosis.[7–9] Lung cancer is more common and mortality higher in areas of high socioeconomic deprivation; it has been estimated that each year, socioeconomic inequalities account for 11 700 excess cases of lung cancer and 9900 potentially avoidable lung cancer deaths in England.[10] High prevalence of smoking, lung comorbid conditions and asbestos exposure, all of which are well-documented risk factors for lung cancer, contribute to high lung cancer incidence and mortality in deprived communities.[11 12]

The presence of lung comorbidity such as chronic obstructive pulmonary disease (COPD) and history of smoking have been associated with a lower likelihood of presenting with lung cancer symptoms early.[13 14] In the lead up to lung cancer diagnosis, vague symptoms may go unnoticed or not considered a legitimate symptom to seek medical attention for, or be misattributed to smoking, ageing or other comorbid conditions such as heart disease or COPD, thereby prolonging help seeking.[5 13 15–26] In addition, stigma attached to lung cancer[23 27–30] and fear of lung cancer diagnosis can deter medical help seeking for symptoms, particularly among smokers,[23–25 29 31–33] leading to advanced stage disease at diagnosis.[19] To date, research has mainly been conducted with patients with lung cancer from a range of socioeconomic groups with varying levels of lung cancer risk, retrospectively exploring the barriers to symptom presentation. Evidence is lacking about how individuals who are at high risk, and without a diagnosis of lung cancer, attribute potential lung cancer symptoms and decide to seek medical help.

Strategies to prompt earlier help seeking for lung cancer symptoms are required. However, evidence is limited regarding optimal methods for promoting earlier presentation through interventions targeted at high risk, highly deprived groups. Mass media[34] and community-based social marketing[35] lung cancer campaigns report limited reach to the most deprived groups. A nurse-led primary care intervention for older adults with a long smoking history or recent cessation reported increased and sustained intentions to seek help with lung cancer symptoms.[36] However, the intervention was not targeted at highly deprived groups. Novel methods to support high-risk groups to engage in early lung cancer diagnosis are required.

The current study used a combination of interviews and focus groups to explore potential barriers to early lung cancer diagnosis and strategies to encourage early help seeking with individuals who are the high risk for lung cancer. Qualitative interviews were used to gain an in-depth understanding of the processes and motivations involved in symptom attribution and medical help seeking for potential lung cancer symptoms in high risk, highly deprived individuals. We targeted socioeconomically deprived areas across three nations of the UK to approach potential participants, and used rigorous sampling procedures to ensure that our sample were high risk for lung cancer. The focus groups were conducted in highly deprived areas with stakeholders who lived or worked in these communities. To our knowledge, this was the first study to explore the influences on early lung cancer diagnosis and intervention preferences targeted at high-risk groups living in the most deprived areas of the UK.

## METHODS

The Consolidated Criteria for Reporting Qualitative Research[37] criteria were used to guide reporting (online supplementary file 1). We used a combination of interviews and focus groups because the interviews were framed around lung health (not lung cancer), whereas the focus groups were framed around preferences for a lung cancer intervention. In addition, key interview findings were presented in the focus groups for consolidation and to facilitate discussion about intervention preferences.

### Participant recruitment and sampling

#### Interviews

Thirty-seven interview participants were recruited through seven primary care general medical practices (GP) in South Wales (Cwm Taf: three practices), England (Liverpool: one practice) and Scotland (Aberdeen: three practices). Using routinely published index of multiple deprivation (IMD) data for England, Scotland and Wales, GP practices with the highest proportion of their patients that reside in the most deprived quintile were contacted. Practice managers were asked to screen GP practice databases purposively for eligible study participants: men and women over the age of 40 years, who were current or former smokers, with a lung condition (COPD including chronic bronchitis and emphysema, interstitial lung disease or occupational lung disease). To overcome methodological limitations associated with retrospective recall, we recruited participants with no previous diagnosis of lung cancer. Participants were initially recruited from GP practices in Cwm Taf, where practice managers were asked to screen databases for current and former smokers, with no parameter set for number of years since quit attempt. Due to an initially high response rate from former smokers in Cwm Taf, subsequent participants in Aberdeen and Liverpool were sampled purposively according to smoking history. One GP practice in Aberdeen was asked to recruit current smokers and recent quitters (within 10 years). Two GP practices in Aberdeen

and one GP practice in Liverpool recruited current smokers only.

To ensure that participants from highly deprived areas were invited to take part in the study, individual postcodes were screened by the research team. Eligible patients from the initial database screen were assigned a pseudo-anonymised participant identifier (PID). PID and postcode were checked against IMD score, and those that resided in the most deprived IMD quintile were eligible for the study. The final list of potentially eligible participants was checked by the GP for ability to provide informed consent, considered by the GP to be a risk to the interviewer or themselves and general health status (e.g. very seriously ill). Participants were excluded if they were diagnosed with lung cancer, were terminally ill or did not have capacity to consent.

## Focus groups

Sixteen participants for the focus groups with members of the public were recruited opportunistically through primary care or local community groups. Participant recruitment through primary care employed the same methods as those used to invite the interview participants. PIDs were checked to ensure that those who took part in the focus groups had not already participated in the interviews. Additional participants were recruited opportunistically through local community respiratory support groups and non-health-related groups in the local community centre. Local community group organisers in areas of high deprivation were contacted and asked for help to recruit members of the public in our target group. Local health service planning groups and health board staff facilitated recruitment of 12 participants for the healthcare professional and community partner focus groups.

## Study procedures

The study received ethical approval from Southampton Central- Hampshire A Research Ethics Committee (16/SC/0589). Written consent and permission to audio-record were obtained on the day of the interviews and focus groups.

## Interviews

Eligible participants were invited by letter with more detailed study information attached, with a reminder at two weeks to non-respondents. Those who returned the study reply slip via a FREEPOST envelope were contacted by the interviewer (GMMcC or JH) to arrange a suitable time and date for the interview, to outline the study and answer any questions.

Interviews were conducted using a semi-structured topic guide to facilitate a discussion about illness perceptions and coping strategies; development was guided by the Common Sense Model[38] (online supplementary file 2). The interview was framed the interview around lung health, rather than lung cancer. The interview aimed to explore experiences of their lung condition, symptom attribution, symptom experience and help seeking behaviour, the influence of smoking history on new or changing symptoms, and if appropriate, lung cancer awareness and beliefs.

A symptom sorting task was used to provide participants with a concrete visual task to increase engagement with the interview in the context of potential low literacy. The task formed a basis for discussion about symptom attribution and experience, where participants were asked to order 11 symptoms from those they would go to the doctor with first, through to the last. The 11 symptoms were selected from the National Institute for Health and Care Excellence guidance for referral of suspected lung cancer (https://www.nice.org.uk/guidance/cg121). The symptoms were re-worded to simplify the language in line with wording found on the National Health Service (NHS) Choices website for lung cancer symptoms and any reference to time scale of symptoms was removed (online supplementary file 2, p. 9). For example, a cough that lasts for three weeks or more was amended to 'persistent cough', and haemoptysis was amended to 'coughing up blood'. The presentation order of the symptoms was rotated between interviews.

To explore potential lung cancer symptom attribution outside of a cancer context, there was no mention of cancer in the interview study information packs or when participants completed the symptom sorting task. If appropriate, participants were asked questions to explore lung cancer awareness and beliefs at the end of the interview or when participants discussed lung cancer unprompted.

Demographic data were collected using a short questionnaire, including three additional measures of socioeconomic group: age, gender, smoking status (quantity and duration), home ownership, occupation and educational attainment. Interviews were conducted until data saturation (no new themes emerging[39]).

## Focus groups

High-risk members of the public and healthcare professionals (e.g. GP, nurse, community pharmacist, community partners) working in areas of deprivation with people with smoking history and/or lung conditions were sent information about the study and invited to take part in focus groups. Focus group participants were explicitly informed that the study was about the development of an intervention about lung cancer. A mutually convenient time, date and location for the focus groups was agreed. The focus groups were conducted using a semi-structured topic guide to explore preferences for an intervention to promote earlier lung cancer diagnosis. Separate topic guides were used for the public and professional groups (online supplementary file 3 and 4). Participants were given a verbal summary of the key findings from the qualitative interviews, and asked to discuss preferences for a potential lung cancer intervention targeted at high-risk, highly deprived individuals. Topics for discussion were: preferred format of an intervention, recommendations

for intervention content, preferred location and facilitator for intervention delivery and recommendations for the inclusion of smoking cessation advice.

## Setting

Most interviews (n=34) took place face-to-face in participant's own homes, with three taking place in a café, local community centre or over the telephone, and lasted between 46 and 146 min (mean 83 min). Family members were present for three interviews but did not participate in the study. Focus groups took place in primary care settings (n=2) or local community centres (n=2). Members of the public who took part in the interviews or focus groups were compensated with a £10 shopping voucher. Healthcare professionals and community partners were not reimbursed for their time.

Interviews and focus groups in England were conducted by JH (PhD), a trained and experienced female qualitative Research Fellow and Medical Sociologist. The Welsh and Scottish interviews and focus groups were conducted by GMMcC (PhD), a female Health Psychologist and trained qualitative Research Associate.

## Data analysis

Interviews and focus groups were audio-recorded and transcribed verbatim. Anonymised transcripts were analysed in detail using the Framework method.[40] Framework analysis is a well-respected and commonly used approach to qualitative data analysis. It was considered particularly suitable for this study due to its transparency and the team work involved.[41] Framework enabled the sharing of synthesised data charts among team members to facilitate participation in analysis and interpretation workshops.

The data were analysed in five stages: familiarisation, identification of a thematic framework, indexing, charting and interpretation. A separate index was created on Microsoft Excel for the interview and focus group data; however, wherever possible, overlap was coded using the same indexing terms, for example, 'barriers to symptom presentation' was commonly discussed in both the interview and focus groups. The index was developed by two researchers (GMMcC and JH). Themes were generated independently and consolidated through discussion in nine interpretation workshops over a nine-month period by GMMcC and JH. The different perspectives of the researchers as noted above was a benefit during analysis and interpretation. Field notes were recorded for each interview and focus group, and incorporated into discussion during the analysis workshops. Although not formally incorporated into the analysis plan, the positioning of each symptom in the attribution task was considered during interpretive workshops. Interpretive themes were generated by JH and GMMcC, and developed with all authors in monthly management meetings. Transcripts and study findings were not checked by participants; however, all participants were mailed a summary of the study findings.

## Patient and public involvement

Patient and public representatives (AMT and GN) were involved in the design of the study and interpretation of study findings in monthly management group meetings. All study materials and topic guides were developed with lay input (AMT and GN) and written to a reading age of 10 years due to potentially low literacy. Reading age was calculated using the Automated Readability Index (www. readabilityformulas.com).

## RESULTS

### Interviews

Of the 397 invited to take part in the study, 78 people returned the study reply slip and declined to participate in the study; reasons for refusal were unknown. Thirty-seven participants agreed to take part in the study. The majority of the sample were female, current smokers and with a mean age of 65 years (table 1). Most had a diagnosis of COPD. All 37 participants resided in the lowest quintile of deprivation for their respective country, of whom 15 were in the most deprived decile. Most participants had left school before age 15 with no formal qualifications, lived in social housing and claimed disability benefit or job seekers allowance.

Key themes were: strategies involved in symptom detection and help seeking behaviour, maintaining short-term health, avoidance of acting on long-term health, the desire to be a model patient and the importance of the relationship with their healthcare professional. See table 2 for illustrative quotes.

### Symptom detection strategies and help seeking

Symptoms discussed during the task were viewed as 'part and parcel' (male, 68, England, current smoker) of their lung condition, other pre-existing comorbidities or smoking habit and were consequently normalised and perceived not to require medical help. Changes to vague or respiratory-type lung cancer symptoms were only taken seriously when remarked on by friends and family or when they impacted on daily life.

Symptoms that could indicate a chest infection were reportedly constantly monitored. Participants discussed using sophisticated strategies such as noticing changes in the colour and consistency of their phlegm or subtle audible changes in their cough to actively detect chest infections. Such strategies were considered important to facilitate early detection and treatment for chest infections through their primary care provider or with rescue packs (emergency packs of steroids and antibiotics that can be kept at home), due to lung condition.

Constant monitoring of phlegm for control of lung condition meant that participants could and would notice haemoptysis, but few reported actively looking for haemoptysis on a regular basis. Disparity between actual and anticipated medical help seeking was reported for haemoptysis. Most participants had not previously experienced haemoptysis, but would anticipate seeking medical

**Table 1** Qualitative interview sample characteristics

| Sample characteristics | Total n=37 |
|---|---|
| **Gender** | |
| Male | 16 |
| Female | 21 |
| **Age, years** | |
| Mean (range) | 64.7 (48–84) |
| **Smoking status** | |
| Current smoker | 18 |
| Occasional smoker | 3 |
| Former smoker, recent quitter (within 5 years) | 5 |
| Former smoker (quit over 5 years ago) | 11 |
| **Deprivation decile** | |
| *Welsh Index of Multiple Deprivation* | |
| Decile 1 (most deprived 10%) | 5 |
| Decile 2 (most deprived 11%–20%) | 10 |
| *Scottish Index of Multiple Deprivation* | |
| Decile 1 (most deprived 10%) | 4 |
| Decile 2 (most deprived 11%–20%) | 12 |
| *English Index of Multiple Deprivation* | |
| Decile 1 (most deprived 10%) | 6 |
| **Self-reported lung condition** | |
| Chronic obstructive pulmonary disease | 26 |
| Chronic bronchitis | 2 |
| Chronic emphysema | 2 |
| Occupational lung disease | 1 |
| Unsure of diagnosis | 4 |
| Missing | 2 |
| **Educational attainment** | |
| Left school at/before age 15 | 29 |
| Completed CSEs, O-Levels or equivalent | 5 |
| Completed A levels or equivalent | 1 |
| Completed further education but not degree | 1 |
| Missing | 1 |
| **Employment** | |
| Employed full-time | 2 |
| Employed part-time | 1 |
| Casual work | 1 |
| Job seekers or disability benefit | 17 |
| Retired | 16 |
| **Home/living arrangement** | |
| Own flat/house | 14 |
| Rent from local authority/housing association | 21 |
| Rent privately | 1 |
| Missing | 1 |

help immediately due to the potentially serious nature of blood. However, some participants who had previously or were currently experiencing haemoptysis attributed

the presence of blood to non-cancer causes such as their stomach ulcer or a previous influenza jab. One participant ascribed the blood in their cough to lung cancer. Some of the participants with experience of haemoptysis did not seek medical help.

### Focus on maintaining short-term health

Participants reported seeking medical help quickly when symptoms were easy to detect, were attributed to what was perceived as a treatable cause and represented an immediate health threat, that is, a chest infection due to lung condition. Participants could often request an appointment the same day as permitted by their GP surgery policies. Prompt help seeking was reportedly due to fear of not being able to breathe and the potentially life-threatening nature of chest infections, and is likely to reflect the need to maintain good health in the short term.

The focus on maintaining short-term health may reflect low general expectations of health, where some participants disclosed surprise at living beyond 60 years of age. In addition, due to fear of potentially hearing bad news, some participants expressed a preference to not ask questions during a consultation or yearly review with the nurse. Participants discussed prioritising day-by-day living over long-term planning, thereby focusing on health in the short term.

### Avoidance of acting on long-term health

Most participants discussed scepticism about the link between lung cancer and smoking. Conversely, participants thought that lung cancer was inevitable due to their current or former lifestyle, including smoking history, working conditions, their lung condition and the reported incidence of lung cancer in their community. For many participants, the topic of lung cancer arose spontaneously. Lung cancer was discussed in the context of perceived inevitability when reflecting on their general lung health and during completion of the symptom task when recalling friends/family with lung cancer. Beliefs about inevitability were often coupled with highly negative fearful and fatalistic beliefs about lung cancer, with no cure and eventual death. Such claims were evidenced by knowing a high proportion of friends and family who were diagnosed with lung cancer and often died. A few participants discussed that a cure for lung cancer involved luck or was 'some miracle' (male, 56, Wales, occasional smoker), reflecting a perceived lack of control over early diagnosis and treatment. Consequently, actual or anticipated medical help seeking for lung cancer symptoms was motivated by pain, or to seek a diagnosis and prognosis to notify family members. However, some participants anticipated refusal of treatment or would even contemplate suicide.

We found differences in how participants with and without dependent family reported responding to symptoms of lung cancer. Female participants with dependent children or grandchildren discussed a motivation to visit the doctor with symptoms suggestive of lung cancer, in

**Table 2**  Illustrative quotes (qualitative interviews)

| Theme | Quote |
|---|---|
| **Symptom detection strategies and help seeking** | |
| Friends and family notice symptoms | *"My daughter might [notice changes to symptoms] cos she mentions it now and then… she'll give me a dig and she'll say 'your breathing's annoying me'. Cos it's heavy breathing so then again there's something wrong".* (Male, 48, Scotland, former smoker) |
| Sophisticated symptom detection strategies/ monitoring of chest infections | *"If [phlegm is] white and bubbly it's not a chest infection. It's only when it goes green so you can tell yourself exactly how close you are to getting an infection… There's just two different kinds of green spittle, if it's fluorescent green then you've got an infection, normal antibiotics won't work with me, if it's the lighter green I'm fine with that one… it's handy to look out for, because you can get the right medication at the right time…because if anything happens to me, there's no one for my kids".* (Female, 48, Scotland, current smoker) |
| Normalisation of haemoptysis | *"Coughing up blood, I do actually get some of that I don't know why, but it could be because of the ulcer thing and that…There again then, well I do get like nosebleeds, and then I'm thinking the blood maybe coming inside and coming down, you swallow it see. So then that will come back up won't it".* (Male, 62, Wales, former smoker) |
| **Focus on maintaining health in the short term** | *"I get worried about having chest infection, I get more worried about today or tomorrow rather than the future. The future that's going ahead for us anyway. Lung cancer's not an issue really".* (Male, 50, Scotland, former smoker) |
| Fear of bad news during a consultation | *"I'm very poor in asking questions cos I don't want to know the results. Simple as that…no I don't ask when they say the oxygen [saturation] is alright I just think well it's alright and it's one thing less I haven't got to worry about".*   (Female, 69, Wales, former smoker) |
| **Avoidance of long-term health outcomes** | |
| Scepticism about the link between smoking and lung cancer | *"You hear occasions where people who don't smoke, who've never smoked. Well how do they get their lung cancer?…I've got [lung cancer] in my head, I'm probably going to get it, if I haven't already got it because of the lifestyle I've had. Where I've worked and everything else, what I've worked with"* . (Male, 68, England, current smoker) |
| Perceived inevitability of    lung cancer/ anticipate  suicide | *"[Lung cancer] is really, really on the forefront on the mind…I just think 'oh god, please don't let me get cancer'…I think if I was to get cancer, I've sometimes said to myself, I'd commit suicide. I would take a pill or something".* (Female, 81, Scotland, current smoker) |
| Avoidance of lung cancer due to social and contextual factors | *"[Lung cancer] worries me but I've got proper problems to worry about [carer for disabled son, problems with social services and benefits claims, insecurity of current council owned housing and problems with area of residence with 'junkies']. I won't worry about it until it's actually here. If I started worrying about eventualities I'd never get anywhere".* (Female, 48, Scotland, current smoker) |
| Lung cancer fatalism/anticipated refusal of treatment | *"Until anything happened and I'm actually told that I've got [lung cancer] , there's nothing I can do about it. I'm really a believer of what's in your cards is already written. So I don't look at anything like that…But if they told me it was cancer, I would go ok then, but I wouldn't take any of the treatments…if it's my time, it's my time. It just doesn't, I don't think I've got any more fight in me for all that. I think that would be the last straw for me. So I just live every day as it comes now, I don't really plan much. So I'm just living in the day, you know. Cos whatever happens, happens anyway".* (Female, 49, Scotland, current smoker) |
| Response to lung cancer symptom/female with dependent family | *"I don't think they can treat [ lung cancer] . You've just got to accept it haven't you…I would go to the doctor [ with a symptom] , I think I would like to know how long I had. Not for me but for [ my son] you know. For him…If it was just me I wouldn't want to know, but because I've got him, [ I would] definitely…When I seen the blood I did think to myself, I flushed it away right away…I seen the blood and I thought no, and I thought I've got to, you know, because of [ my son]. The only way I would want to know is because of him. If I was by myself I would just say, don't want to know…Can't just think about myself I've got to think about him as well".* (Female, 68, Scotland, current smoker) |
| **The model patient** | |
| Perception of healthcare professionals attitude to smokers | *"You feel as though you're an alien because you smoke, you feel as so they just look at you and say 'urghh', you know".* (Female, 52, Scotland, current smoker) |
| Critical of people who waste National Health Service (NHS) resources | *"I can guarantee if I went this Monday and go next Monday the same people are sitting there. I'm being honest, they're a drain on society on the NHS, but that's the way they live…these people that go there are not really ill, I think they're just seeking attention".* (Male, 78, England, current smoker) |
| **Relationship with healthcare professional** | |
| Disclosure of highly sensitive personal problem | *"Some people are friendly and not stony faced…if (the HCP) can't even start a conversation with the simplest of ice breakers then how can people tell about pooping themselves when they're coughing up".* (Female, 48, Scotland, current smoker) |

Continued

| Table 2 | Continued |
|---|---|
| **Theme** | **Quote** |
| Good relationship with GP | *"I'm alright with (one GP), you could tell her anything, I've shocked her sometimes".* (Female, 51, England, current smoker) |

order to receive a prognosis to enable childcare arrangements after death. Women with dependent children who held more positive beliefs about lung cancer treatment reported the need to seek help for treatment to 'stay healthy' and prolong life. Participants with no dependent family were more likely to ignore lung cancer symptoms, or anticipate seeking medical help if in pain but refuse treatment.

### The model patient

Participants discussed a sense of lack of entitlement to health services due to smoking habit, where respiratory-type symptoms of lung cancer were perceived as self-inflicted. For some, this was reinforced by an actual or expected 'smoking lecture' each time they sought help from healthcare professionals; the lecture made participants feel ostracised, particularly when smoking was used as a coping mechanism and contributed to not feeling worthy of seeking medical help. Some participants perceived that they may be treated differently by health professionals because they live in an area of deprivation, and discussed a potential power imbalance during consultations.

Conversely, participants reported high criticism towards people who were perceived to waste, exploit and overuse NHS resources. They cited drug addicts, illegitimate benefits claimers, older people wanting social interaction and people with coughs and colds as over users of the health service. Such beliefs may reflect a downward comparison to other more stigmatised service users to legitimise their own help seeking. In order to be considered a model and non-problem patient, participants discussed legitimising their own help seeking by only consulting when absolutely necessary—and often after trying their 'own cures', that is, cough medicine from the pharmacist—to not burden the doctors. Infrequent attenders or 'good service users' discussed feeling a sense of superiority for being a model patient.

### Relationship with the healthcare professional

Some participants disclosed traumatic events in their lives including physical and sexual abuse, leading to tobacco dependence and alcohol addiction. In addition, more than half of the sample described symptoms of depression and anxiety. Therefore, the reported relationship with their healthcare professional was important when considering whether to present with lung symptoms. Participants discussed the need to feel understood and not judged by their healthcare professional, with their personal history taken into account in the context of health behaviour such as smoking.

Those who discussed feeling comfortable, safe and not judged by their chosen healthcare professional felt encouraged to present with symptoms. Some participants reported that they were prepared to wait up to three weeks for an appointment with their preferred healthcare professional to discuss worrisome and potentially serious symptoms that could indicate lung cancer. Many participants reported problems with maintaining continuity of care, highlighting problems with the stretched NHS.

### Focus groups

Two public focus groups were conducted in Wales and England. Most participants were female and former smokers, and all participants were diagnosed with a lung condition. Two professional focus groups were conducted in Wales. Most participants were female, and were medical professionals (table 3).

Key themes discussed were: barriers to early lung cancer diagnosis, and preferences regarding the format and content of an intervention for the early detection of lung cancer. See table 4 for illustrative quotes.

### Barriers to lung cancer symptom presentation

The public and stakeholder focus groups confirmed our interview findings, where fear of wasting the doctor's time with trivial symptoms and fear of being judged or lectured about smoking was perceived to deter medical help seeking for potential lung cancer symptoms. In addition, the health professional group supported our findings that patients with lung conditions tend to be preoccupied by chest infections. However, we found potential disparity between the patient-reported experience of the GP's approach to smoking and the healthcare professional reported approach to smoking cessation. Healthcare professionals in Wales discussed new guidance that discourages health professionals from 'lecturing' patients, suggesting the patient-reported experience may be based on previous healthcare interactions, and they consequently anticipate a lecture. Alternatively, healthcare professionals may be unaware of new guidance, or not adhere to new guidance and consequently continue to 'lecture' patients about smoking.

### Potential format of an intervention to support earlier lung cancer diagnosis

All groups discussed a preference for community based interventions, away from a traditional healthcare setting, for example, a community event, talk in a community venue or health check bus, similar to breast screening mobile units. The anonymous and relaxed nature of such an intervention meant that intervention participants

**Table 3** Focus group characteristics

| Members of the public | N participants | Healthcare professionals and community partners | N participants |
|---|---|---|---|
| **Group 1, England** | **Total n=7** | **Group 3, Wales** | **Total n=5** |
| *Gender* | | *Gender* | |
| Female | 6 | Female | 2 |
| Male | 1 | Male | 3 |
| *Smoking status* | | *Occupation* | |
| Current smoker | 3 | Community nurse | 1 |
| Former smoker | 3 | Support group facilitator | 1 |
| Never smoker | 1 | Community partner | 1 |
| *Self-reported lung condition* | | Third sector representative | 1 |
| Chronic obstructive pulmonary disease (COPD) | 7 | Public health representative | 1 |
| Recruited through primary care and community groups | | Recruited through the health board | |
| **Group 2, Wales** | **Total n=9** | **Group 4, Wales** | **Total n=7** |
| *Gender* | | *Gender* | |
| Female | 5 | Female | 6 |
| Male | 4 | Male | 1 |
| *Smoking status* | | *Occupation* | |
| Current smoker | 3 | Practice manager | 1 |
| Former smoker | 4 | Pharmacist | 1 |
| Never smoker | 2 | General practitioner | 2 |
| *Self-reported lung condition* | | Practice nurse | 2 |
| COPD | 9 | Medical student | 1 |
| Recruited through community groups | | Recruited through the health board/ primary care | |

would feel they were not wasting GP time; rather it would act as a signal that their attendance at the event was desired. Participants compared this with a visit to the doctor, where they discussed a feeling of wasting the GP's time because they were not invited to attend. It was considered important that the intervention facilitator was knowledgeable or trained, non-judgemental, easy to talk to and approachable, highlighting the importance of relational aspects of a lung cancer intervention. Participants suggested a nurse, pharmacist, trained patient representative or community worker.

### Intervention content
The public groups requested more information about the symptoms of lung cancer. However, the healthcare professional groups felt that current lung cancer symptom information was too broad, leading to dismissal and potential avoidance of lung cancer information because people with smoking history or comorbid lung conditions experience most of the symptoms daily. To overcome this problem, the healthcare professionals groups discussed the need for more specific symptom information, emphasising changes to normal symptoms and coupled with information about risk factors for lung cancer.

To modify negative beliefs about lung cancer, the health professionals groups suggested using positive stories to communicate messages about the importance of lung cancer early diagnosis and highlight the potential for survival outcomes with early stage detection.

The inclusion of smoking cessation information in a lung cancer intervention was considered important by all groups. However, the manner in which smoking cessation could be approached was discussed as key to effective promotion of smoking cessation. Participants suggested highlighting the benefits of stopping smoking in a gentle and relaxed manner to encourage choice to quit.

### DISCUSSION
Our study was the first to explore the influences on lung cancer symptom presentation in high risk, highly deprived groups across three nations of the UK. Preferences for an intervention targeted at high-risk groups were ascertained through focus groups. We found evidence from the interviews and focus groups that individuals who are at high risk for lung cancer tend to be preoccupied by maintaining health in the short term. Prioritising the

**Table 4** Illustrative quotes (focus groups)

| Theme | Quote |
|---|---|
| **Barriers to lung cancer symptom presentation** | |
| Fixation on chest infections | *"People tend to be fixated on a [chest] infection and they want their next rescue pack ready cos almost as if it's inevitable; it's going to happen in the next month or so".* (Focus group 4) |
| Difference in perception around healthcare professional approach to patients' smoking | *"I think there is a gulf between what people believe their GP would say to them if they do actually talk about [smoking] as opposed to what that conversation actually is in reality….But certainly as far as the formal training coming out of public health, if they are doing that then there is, that's not a lecture…But that's what people fear is going to be what they're going to be told".* (Focus group 3) |
| **Potential format of an intervention to support earlier lung cancer diagnosis** | *Participant 2: "So what I'm saying is, you know them mobile buses… in the shopping area, where people go shops, or outside the hospital… So they set them up and people are walking past, and even though they can't be bothered to go to the doctors, and they look and they think I'll just pop in".*<br>*Participant 1: "Cos you wouldn't hesitate you know, you'd just go in".*<br>*Participant 2: "You're just a person, they don't know and they're just seeing what's there, or what's there or what's the problem with you. If there's no problem".*<br>*Participant 3: "People think you don't want to think you're, feel as if you're wasting the doctor's time".* (Focus group 1) |
| **Intervention content** | |
| More specific symptom advice | *Participant 1: "Yeah I think when you say 'cough' it's a bit broad and it's a bit…You know, you've had a cough for 2 weeks , off you go".*<br>*Participant 3: "It'd be useful if it was a change in your regular cough".* (Focus group 4) |
| Messages to combat negative beliefs | *"Positive messages, particularly around lung cancer because everybody, you know it's like a death knell isn't it? And actually it's not, it doesn't have to be. You know you're talking here about early diagnosis which is a big deal isn't it".* (Focus group 3) |
| Smoking cessation | *"You've got to include [smoking cessation information] …I think it's how you deliver the message…not in such a way you feel ashamed for smoking. I've noticed [ the nurse] has got a way of telling patients how to stop smoking, she does it in a, not in a 'well you should stop smoking', that kind of way. She'll say 'have you ever thought about giving it up. You know it would improve your chest a bit'. And I've seen [the nurse do it] more in a non-lecturey basis, more of a, 'have you ever thought about it?' Relaxed, warmer manner. So I'm not lecturing you, it's your choice. You know it's bad for you".* (Focus group 4) |

daily management of their lung condition led to avoiding consideration of long-term health problems such as lung cancer, to gain a sense of control over health in the context of difficult personal circumstances. Health beliefs were found to underpin behaviour in relation to medical help seeking, where perceptions of 'inevitable but curable' chest infections led to immediate help seeking. However, 'inevitable but incurable' lung cancer led to inaction when faced with potentially serious symptoms and anticipated refusal of treatment. Interview participants felt that the relationship with the healthcare professional was key when considering medical help seeking. The importance of the relational interaction between provider and patient was mirrored in the focus groups, where participants felt that a non-judgemental intervention facilitator was important. Multifaceted community-based interventions, away from the traditional healthcare setting, were preferred by participants.

Previous empirical studies report prolonged lung cancer symptom presentation due to misattribution[5 13 15–26 33 42] and in our study, we found evidence that participants normalised their symptoms indicative of lung cancer to smoking habit, and lung and other comorbid conditions. In contrast to previous studies that report haemoptysis as a facilitator to prompt medical help seeking,[13 25 27 43–45] current participants with experience of haemoptysis

reported described avoidant coping, and normalisation when blood was noticed. Dismissal and normalisation of haemoptysis may be specific to socioeconomically deprived groups. Our highly deprived sample reported daily struggles with complex physical and mental health needs, and with the challenges associated with living on no or limited income. Previous studies in socioeconomically deprived communities report that in the context of competing life demands, health was dealt with reactively and with low priority.[46 47]

Fear of being ineligible for treatment due to lifestyle has not been well described in studies with patients with lung cancer or those at high risk.[44 48] In contrast, participants in the current study described feeling disentitled to medical services in the context of their lifestyle and circumstances. The underlying concept of health service candidacy (perceived eligibility for healthcare)[47] may explain why participants felt unworthy of seeking medical help and is likely to be of particular importance in our highly deprived sample. In addition to challenging life circumstances, interview and focus group participants reported fear of being judged and ignored by health professionals due to their smoking habit or perceived social standing, contributing to feelings of unworthiness. Participants reported the desire to be a model patient and to not waste valuable GP time, which influenced medical

help seeking. Although the desire to be a 'good citizen' has previously been reported,[24 25] to our knowledge, the current study was the first to explore perceptions of appropriate consultation behaviour in a highly deprived sample. Our emerging findings related to candidacy, combined with the desire to exhibit 'good' consultation behaviour, may contribute to normalisation of symptoms previously regarded as serious and therefore discourage help seeking. Consequently, disadvantaged populations are likely to focus on health in the short term, and ignore long-term health issues which may lead to advanced stage lung cancer diagnosis.

We found that participants held seemingly contradictory views on their lung cancer susceptibility, reporting scepticism about the causal role of smoking in lung cancer alongside perceived inevitability of lung cancer. Beliefs about the link between smoking and lung cancer may reflect societal stigma towards smoking, where participants downplay the negative effects of smoking, possibly to legitimise medical help seeking for symptoms considered related to smoking. Perceived inevitability of lung cancer is likely to reflect high levels of exposure in social networks where there is high incidence and poor outcomes of lung cancer,[10] which should minimise normalisation of lung cancer symptoms and prompt help seeking.[49] Contrary to previous studies, current participants reported feeling that lung cancer was inevitable while simultaneously normalising and ignoring haemoptysis, possibly due to a combination of high fear and fatalism about lung cancer, difficult life circumstances and low perceived health service candidacy. High-risk individuals who believe that they cannot legitimately seek medical help because of their former or current lifestyle may therefore be resigned to the prospect of developing lung cancer.

A major strength of this study was the rigorous sampling procedure. We screened postcodes to ensure participants resided in the lowest quintile of deprivation, and measured multiple additional indicators of deprivation. Individual and area level indicators confirm that our sample was highly deprived, for instance, most were unemployed and seeking benefits, and rented social housing. In addition, we recruited participants with no previous diagnosis of lung cancer, without mention of lung cancer until discussed by participants during the interview, or at the end of the interview. These recruitment and interview procedures meant we were able to explore previous and anticipated lung cancer symptom presentation in those who were symptomatic or asymptomatic. This strategy was employed to overcome the methodological limitations associated with studying either retrospective or anticipated symptom presentation in isolation[50]. However, our qualitative study was unable to establish causal links between barriers and help seeking, nor can we generalise or compare the findings to high socioeconomic groups; instead, we conducted an in-depth study to explore how best to engage high-risk, highly deprived individuals in early lung cancer diagnosis. Although we carefully sampled participants and collected additional demographic measures to validate our sampling frame, some GP practices were asked to recruit by specific smoking status rather than the whole range of smoking status, potentially introducing bias to our sample. In addition, we were unable to conduct a focus group in Scotland due to low response, which is a potential limitation of the study. Finally, focus group participants were recruited opportunistically, with the potential that participants were more favourably disposed to an intervention.

### Practice and policy implications

With a comorbid lung condition and smoking history, those who are high risk for lung cancer will, in the main, be symptomatic. To avoid normalisation of symptoms, it is important to highlight the significance of changing and multiple symptoms. High-risk individuals should be empowered to seek timely medical help and made to feel welcome, not judged or blamed for their current or former lifestyle. For instance, interventions targeted at disadvantaged populations could be conducted outside of the traditional healthcare setting. Our findings highlight the importance of an intervention where participants would be invited to attend, as opposed to presenting to the GP surgery, in order to eliminate concerns about wasting GP time and legitimise their attendance. Community-based interventions have the potential to harness the relational aspects of help seeking, through interventions led by non-judgemental and welcoming facilitators. It is possible that previous mass media and social marketing lung cancer awareness interventions report low campaign reach to deprived groups[34 35] in part because they were not designed to motivate help seeking through intensive approaches to build trusting relationships and confidence. More research is required to understand how the relational aspects of help seeking could be operationalised in an intervention.

Over half of the current sample described mental health problems and/or difficult current or former life circumstances. Intervention developers and healthcare professionals in highly deprived communities should be aware of these wider social and contextual factors; they should receive training to recognise such circumstances and know how to appropriately signpost. Finally, we suggest that the current UK health system may encourage patients with a lung condition to focus on short-term management of their condition. GP prescribing of antibiotics and the use of rescue packs (prescribed antibiotics for storage at home in the event of an exacerbation) may inadvertently reinforce patients to detect and act on symptoms of a chest infection.[9] There is potential that this current standard of care could be adapted to educate and encourage patients with a lung condition to detect symptoms of lung cancer, thereby shifting the focus to long-term health. More research is required to understand how to motivate highly deprived groups to consider health in the long term, while recognising the wider social determinants of health.[51]

## CONCLUSION

The challenges of living in an area of deprivation with social exclusion issues, combined with fear of judgement by health professionals, contribute to avoidance and ignoring of lung cancer symptoms. Multi-faceted community based interventions are required to highlight lung cancer symptoms, the importance of early diagnosis and empower people who are at high risk for lung cancer to seek timely medical help.

**Author affiliations**
[1]Division of Population Medicine, Cardiff University, Cardiff, UK
[2]North Wales Centre for Primary Care Research, Bangor University, Wrexham, UK
[3]Centre for Trials Research, School of Medicine, Cardiff University, Cardiff, UK
[4]Division of Applied Health Science, University of Aberdeen, Aberdeen, UK
[5]Leeds Institute of Health Sciences, University of Leeds, Leeds, UK
[6]Division of Population Medicine, Patient and Public Involvement, Cardiff University, Cardiff, UK
[7]Cwm Taf Morgannwg Public Health Team, Public Health Wales, Merthyr Tydfil, UK
[8]Division of Population Medicine, School of Medicine, Cardiff University, Cardiff, UK

**Acknowledgements** The authors would like to thank everyone who helped with recruitment for this study, and all the participants who took part in the study; without these individuals this study would not have been possible. The authors would also like to thank the Advisory Group (Anthony Byrne, Gareth Collier, Adrian Edwards, Dyfed Huws, David Lewis, Maura Matthews and Fiona Walter) and the study Administrator Lucy Watkins, for their ongoing support and advice throughout the study.

**Contributors** All authors designed the study. GMcC and JH conducted, coded and analysed the interviews and focus group data. All authors contributed to the interpretation of data. GMcC drafted the manuscript and all authors contributed to the review and editing of the manuscript. All authors read and approved the final manuscript.

**Funding** This work was supported by Cancer Research UK (grant reference number: C16377/A22034).

**Competing interests** None declared.

**Patient consent for publication** Not required.

**Ethics approval** The study received ethical approval from Southampton Central-Hampshire A Research Ethics Committee (16/SC/0589).

**Provenance and peer review** Not commissioned; externally peer reviewed.

**Data sharing statement** The data are anonymised transcripts. Requests to share data will be considered by the PI on an individual basis in line with a data sharing agreement and subject to compliance with Cardiff University regulations whilst ensuring no loss of confidentiality on the part of the study participants.

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
