## [Reviewer comments · BMJ Open]

ARTICLE DETAILS

TITLE (PROVISIONAL)	Engaging high-risk groups in early lung cancer diagnosis: a qualitative study of symptom presentation and intervention preferences amongst the UK's most deprived communities
AUTHORS	McCutchan, GM; Hiscock, Julia; Hood, Kerenza; Murchie, Peter; Neal, Richard; Newton, Gareth; Thomas, Sara; Thomas, Ann Maria; Brain, Kate

VERSION 1 - REVIEW

REVIEWER	Julia Mueller University of Manchester, United Kingdom
REVIEW RETURNED	05-Sep-2018

GENERAL COMMENTS	Thank you for the opportunity to review this interesting study. Overall, this study used appropriate methodology to address the aims. The methods, findings and the relevance and significance of the findings are presented clearly. I have included some comments below which I think will help to improve the manuscript. p. 3 Lines 4-5: "This was the first study to use in-depth qualitative methods to explore how to engage individuals who are highest risk for lung cancer in early lung cancer detection." Some work in this field has been done previously – see e.g. Smith et al, "Developing a complex intervention to reduce time to presentation with symptoms of lung cancer", where qualitative methods were also used to explore views of those at high risk of lung cancer to develop an intervention for earlier diagnosis. I think the novelty of the study lies in the engagement of at-risk individuals from socioeconomically deprived backgrounds, and this should be emphasised here. p. 5, line 13-15: "GP practices with the highest proportion of their patients that reside in the lowest quintile of deprivation were contacted." How was this determined, using what data source? p. 5, lines 19-22: "GP practices in Cwm Taf were asked to recruit current and former smokers, with no parameter on number of years since quit attempt. One GP practice in Aberdeen was asked to recruit current smokers and recent quitters (within ten years). Two GP practices in Aberdeen and one GP practice in Liverpool recruited current smokers only." It seems a bit odd that individual GP practices recruited specific smoking status only, instead of each GP practice recruiting the whole range. I suspect this may be due to logistical/practical considerations? I think some more explanation could help here.
--

Also, is there a potential for bias due to this recruitment method?
May be worth discussing this in the limitations section.

p. 6, lines 17-20: How did participants consent? Did you attach a stamped return envelope?

p. 7, lines 15-23: In the 'Participant recruitment and sampling' section, you state that high-risk patients were included in focus groups, but in the 'Study procedures' section this is not quite clear, as you only mention how healthcare professionals and community partners were contacted.

p. 8, line 29: Can you specify which reading level test you used (e.g. Flesch Kincaid)?

p. 10, line 6: Figure 1 and Figure 2 are tables. I suggest renaming this to avoid confusion.

The results section would flow better and would be easier to follow if quotes were incorporated into the text rather than a separate table.

Throughout the results section, you should make it clearer that accounts of events/actions are self-reported by participants, rather than necessarily reflecting an objective reality. For example, "Symptoms that could indicate a chest infection were constantly monitored"; suggest changing to "Symptoms that could indicate a chest infection were reportedly constantly monitored". I suggest making this clearer throughout the results and discussion, as participants' accounts may be subject to various forms of bias (e.g. social desirability, recall...).

p. 14 line 14-17: "Healthcare professionals in Wales discussed new guidance that discourages health professionals from 'lecturing' patients, suggesting the patient reported experience may be based on previous healthcare interactions, and they consequently anticipate a lecture."

Another possibility is that not all healthcare professionals are aware of or adhere to new guidance and continue to 'lecture' their patients. Based on self-reports alone, we cannot be sure which perception is more likely to reflect reality.

Discussion:

p. 16, lines 9-11: "These recruitment and interview procedures meant we were able to explore actual and anticipated lung cancer symptom presentation in those who were symptomatic or asymptomatic."

I would argue that this study does not allow an exploration of "actual" help-seeking, but rather accesses this through participants' self-reports. I would therefore suggest re-phrasing this, for example: "we were able to explore participants' accounts of previous and anticipated lung cancer symptom presentation"

p. 17, line 18-19: "as a consequence of high lung cancer incidence in areas of deprivation."

A reference is needed here to evidence the high incidence of lung cancer in deprived areas. Also, do you mean incidence or prevalence? I suggest prevalence may be the more appropriate term here.

p. 16, lines 4-17: The limitations of qualitative research should also briefly be mentioned here. While the methodology is appropriate to the research aim, causal links between factors (e.g. fear/denial and delays to help-seeking) cannot be established using this method. Furthermore, this methodology cannot establish whether certain processes are more common among deprived populations than non-deprived populations (e.g. normalisation of haemoptysis). Thus while the method appropriate it is worth mentioning the limitations.

p. 16 line 31

You discuss that some participants felt they may not be eligible for treatment due to lifestyle. Corner et al (“Is late diagnosis of lung cancer inevitable? Interview study of patients’ recollections of symptoms before diagnosis”) report a similar finding, although only one participant in their study mentioned this. If more participants in your study discussed this, then you have a) found further evidence that this theme is of importance and b) it may be an indication that it is of particular importance among deprived groups (this would need to be confirmed using larger-scale, quantitative methods though). I suggest discussing this existing evidence here.

p. 17, lines 20-22: “However, we report normalisation and ignoring of haemoptysis, possibly due to a combination of high fear and fatalism of lung cancer, difficult life circumstances and low perceived health service Candidacy.”

A similar finding (i.e. that fear of cancer and fatalism may contribute to delays) was found by Tod et al (“Diagnostic delay in lung cancer: a qualitative study”). However, other studies (e.g. Corner et al, “Experience of health changes and reasons for delay in seeking care: A UK study of the months prior to the diagnosis of lung cancer”) have found no obvious signs of fear or denial. Corner et al report that their participants seemed to minimise and normalise symptoms, but found no evidence that participants suspected lung cancer and delayed out of fear. It would be good to discuss your finding in light of this existing evidence. How is your study different, and what might account for these different findings?

p. 18, lines 10-13: “Finally, the current UK health system potentially encourages patients with a lung condition to focus on short term management of their condition. GP prescribing of antibiotics and the use of rescue packs may inadvertently reinforce patients to detect and act on symptoms of a chest infection.”

I think a little more explanation and some referencing is needed here. Are these known problems, and is there evidence in the literature that certain GP behaviours reinforce focus on short-term management?

p.17/18: The section on practice and policy implications could be improved by discussing your findings in light of existing interventions to encourage earlier diagnosis for lung cancer. There have been various community-based public awareness-raising interventions as well as GP-based interventions (e.g. Be Clear on Cancer, 3 week cough campaign...). How do your findings add to these? Do existing interventions address the issues identified by your study? How should existing interventions be adapted, according to your findings?

Minor comments:

Abstract

	p. 2 Line 12: Word “in” missing (GP practices in socioeconomically deprived areas...) p. 2 Line 14-16: Sentence is incomplete (Four focus groups with high-risk individuals and local stakeholders (healthcare professionals and community partners) [were conducted] to explore preferences for an intervention to promote early lung cancer symptom presentation.) p. 4, line 34: To ensure that our sample WAS at the highest risk of lung cancer p. 11, lines 7-9: “When symptoms were easy to detect, they were attributed to what was perceived as a treatable cause and represented an immediate health threat i.e. a chest infection due to lung condition, participants sought medical help quickly” There seems to be a word missing in the sentence (...lung condition, AND participants sought medical help quickly. ?) p. 12, line 22 – p. 13, line 2: “Therefore, the relationship with their healthcare professional was important when considering whether to present with lung symptoms; to feel understood by the healthcare professional, with their personal history taken into account in the context of health behaviour such as smoking.” This sentence is very long and a bit awkward to read – suggest revising to improve the reading flow. p. 16, line 4: “A major strength of this study was the rigorous sampling procedures.” Should be either “were the rigorous sampling procedures” or “was the rigorous sampling procedure”.
--	--

REVIEWER	Professor Michael Peake University College London Hospitals and University of Leicester; UK.
REVIEW RETURNED	07-Sep-2018

GENERAL COMMENTS	This is a very well designed and conducted study addressing a very relevant topic. Its findings will be of significant value in the design of future public awareness campaigns and in facilitating future research in the field. It is clear and well written with very little use of unnecessary jargon. It has particular value in that it points to multiple ways in which better engagement with and impact on, ‘hard-to-reach’ groups could be better effected. For example, the findings supports the view that a change in negative attitudes to the benefits and value of early diagnosis are likely to be as important in any campaign as raising awareness of symptoms; it adds weight to the finding of the ICBP about UK patients being inhibited from attending their GP for fear of wasting their time; it highlights the value of continuity of care by a single GP who, the patient knows and trusts; it emphasises the potential value of allowing people to seek health care advice in a more anonymous setting outside that of their normal General Practice. I have no significant comments or suggested changes except for some relatively minor comments on parts of the introduction:
--

	Introduction page 4, line 2 Latest 5 yr survival figure for England from ONS (Cancer survival in England: adult, stage at diagnosis and childhood – patients followed up to 2016) for patients diagnosed in 2011 was 14% for males and 17,5% for women. Introduction page 4, lines 16-17 – mining is cited as an occupational cause of lung cancer. In a UK context this would be presumed to refer to coal mining and the evidence of an association between coal mining and lung cancer is very limited, with a much greater association being seen in quarrymen. The reference given by the authors for this is Malhotra (ref 12) and his study did not refer specifically to coal mining in the UK except where there is true silicosis. So, I think [coal] mining is a bad example to use and would suggest referring to asbestos exposure (using the same reference). Introduction page 4, line 23 – using the word ‘emphysema’ here is inappropriate since emphysema is one feature of COPD and there is no logic for referring to this specific, and essentially pathological, feature of COPD. So, I suggest simply changing the word emphysema to COPD.
--	--

REVIEWER	Dr Afrodita Marcu University of Surrey, UK
REVIEW RETURNED	10-Sep-2018

GENERAL COMMENTS	I enjoyed reading this paper about early lung cancer detection in high-risk groups in UK's most deprived communities. This is a well-written paper with an original study design and insightful analysis, however it needs greater clarification regarding the methods of data collection and analysis and it would benefit from a refining of the discussion. Please see my detailed feedback and recommendations below. Abstract While the Abstract is overall well-written, it could benefit from some details for greater clarity: how many patients were recruited in the interviews vs. the focus groups? How many healthcare professionals and community partners? It needs to be made clear that the participants were cancer-free and that the symptom attribution task explored help-seeking intentions. The conclusions could be phrased more tentatively, e.g. ‘community-based interventions might be a way to empower this population...’ Introduction The Introduction draws on pertinent literature relating to lung cancer, delayed presentation and socio-economic inequalities, and fully justify the need to address what might motivate early symptomatic presentation among deprived groups. However, a justification is needed for why interventions are seen as the solution to delayed help-seeking and what these should consist of. The authors should include examples of interventions which have been developed to promote timely symptomatic presentation among high-risk individuals. The authors should clarify if there is an evidence gap around the patient factors that enable early
--

presentation and include a stronger rationale for why a qualitative approach was undertaken and how it was expected to address this gap. The aim of the study could be phrased in ways that reflect the strengths of qualitative approaches, e.g. 'This qualitative study aimed to understand the sense-making and motivations underpinning early presentation...' or 'what would motivate early presentation... and acceptance of interventions promoting early presentation.'

Method

Greater clarity is needed as to how many interviews and focus groups, respectively, were conducted, and with how many participants. Was cancer made explicit, and were any stimulus materials used in the focus groups? Under 'interviews', 7 high-risk individuals were recruited, yet 37 interviews were carried out. It would have been useful to include the topic guides for the interviews and the focus groups, respectively, as an appendix. The focus groups feel quite disjointed from the interviews, and there needs to be a clearer rationale for why the 37 interviews did not explore the content and format of potential interventions, yet separate focus groups did.

The choice of Framework Analysis needs to be better justified and aligned with the qualitative approach and aims of the study, which seem rather exploratory in nature as presented in the Introduction. The use of the Common-Sense Model of Illness to conduct the interviews, on the other hand, suggests a rather deductive approach, hence the findings, using the Framework approach, could have been mapped onto the key dimensions of the model. The concept of data saturation, which originated in Grounded Theory, may not be appropriate within the Framework Analysis approach – the authors need to check this and rephrase as needed.

The analytic approach to the sorting task needs to be made more explicit and the authors need to report the results (or explain why these results are not included in the paper).

Results and Discussion

The developed themes reflect well the quotes included, however the description of the findings needs to be more attuned to the qualitative paradigm. Phrases like "disparity between actual and anticipated medical help seeking was found for haemoptysis" and "some participants who had previously or were currently experiencing haemoptysis reported normalisation, leading to delays in medical help seeking or no help seeking" are rather unclear: had any participants sought help for symptoms indicative of lung cancer? How many had previously presented with lung cancer symptoms? How was 'delay' explored during the interview? It would have been interesting to know whether the high-risk participants spontaneously mentioned lung cancer and under what circumstances (given that the researchers avoided mentioning 'cancer' in the interviews).

Lines 20-21, p. 16: if the participants in this study were cancer-free, they could not have 'normalised their lung cancer symptoms' – perhaps rephrase as 'their symptoms indicative of lung cancer'.

	The assertion that this study was “the first to explore the influences on lung cancer symptom presentation and intervention preferences in high risk, highly deprived groups” is somehow not warranted given that intervention preferences were not explored during the 37 interviews with high-risk individuals. It feels like a missed opportunity that the useful concept of candidacy (Dixon-Woods et al., 2006) was not introduced earlier in the paper in relation to how people from low socioeconomic backgrounds seek medical help, or in the development of the themes. This would have made the themes more interpretative and would have added more theoretical layers to the analysis. Minor comments: Line 12 in the Introduction: it would be more accurate to say that socioeconomic inequalities increase the likelihood of late presentation rather than the likelihood of a symptomatic individual to have lung cancer. Lines 21-24 in the Introduction: the misattribution of vague symptoms may not be itself a cause of delayed help-seeking, but the fact that such symptoms are not considered legitimate reasons to seek medical attention. Line 31 in the Introduction: given that the study was conducted with cancer-free participants, phrase the aims as exploring what might motivate early symptomatic presentation. Perhaps replace the term ‘fixate’ with ‘prioritize’ or ‘being preoccupied by’, as fixation implies obsession. Suggest moving the paragraph on the strengths and weaknesses of the study towards the end of the Discussion.
--	---

VERSION 1 – AUTHOR RESPONSE

Reviewer 1

Thank you for the opportunity to review this interesting study. Overall, this study used appropriate methodology to address the aims. The methods, findings and the relevance and significance of the findings are presented clearly. I have included some comments below which I think will help to improve the manuscript.

(1) Reviewer’s comments: p. 3 Lines 4-5: “This was the first study to use in-depth qualitative methods to explore how to engage individuals who are highest risk for lung cancer in early lung cancer detection.” Some work in this field has been done previously – see e.g. Smith et al, “Developing a complex intervention to reduce time to presentation with symptoms of lung cancer”, where qualitative methods were also used to explore views of those at high risk of lung cancer to develop an intervention for earlier diagnosis. I think the novelty of the study lies in the engagement of at-risk individuals from socioeconomically deprived backgrounds, and this should be emphasised here.

Authors’ response: Thank you for your comment. The wording has been amended to emphasise that the sample were high risk and highly deprived: “This was the first study to use in-depth qualitative methods to explore how to engage high risk individuals from socioeconomically deprived backgrounds in early lung cancer detection” (page 3, lines 4-5).

(2) Reviewer’s comments: p. 5, line 13-15: “GP practices with the highest proportion of their patients that reside in the lowest quintile of deprivation were contacted.” How was this determined, using what data source?

Authors' response: Index of Multiple Deprivation data was used to determine the percentage of patients in each practice who reside in the most deprived quintile. We selected GP practices that had the highest proportion of their practice population living in the lowest quintile of deprivation. This point has been clarified on page 5, lines 31-33: "Using routinely published index of multiple deprivation (IMD) data for England, Scotland and Wales, GP practices with the highest proportion of their patients that reside in the most deprived quintile were contacted."

(3) Reviewer's comments: p. 5, lines 19-22: "GP practices in Cwm Taf were asked to recruit current and former smokers, with no parameter on number of years since quit attempt. One GP practice in Aberdeen was asked to recruit current smokers and recent quitters (within ten years). Two GP practices in Aberdeen and one GP practice in Liverpool recruited current smokers only." It seems a bit odd that individual GP practices recruited specific smoking status only, instead of each GP practice recruiting the whole range. I suspect this may be due to logistical/practical considerations? I think some more explanation could help here. Also, is there a potential for bias due to this recruitment method? May be worth discussing this in the limitations section.

Authors' response: Participant recruitment for the interviews was staged due to local approvals of the study at Health Board level. Participants were first recruited from Cwm Taf (South Wales), then Grampian (Scotland) and finally, Liverpool (England). We aimed to invite both current and former smokers from all practices; however, when recruitment started in Cwm Taf (the first site), the highest response was from former smokers who quit many years ago. Therefore, practices that were recruited after the initial interviews were asked to recruit current smokers/recent quitters. We asked practices to recruit by specific smoking status because the researcher was unable to obtain smoking status before the interview; we anticipated that asking participants to indicate smoking status on their study reply slip would deter participation in the study.

The wording has been amended to reflect this point: "Participants were initially recruited from GP practices in Cwm Taf, where practice managers were asked to screen databases for current and former smokers, with no parameter set for number of years since quit attempt. Due to an initially high response rate from former smokers in Cwm Taf, subsequent participants in Aberdeen and Liverpool were sampled purposively according to smoking history..." (Page 6, lines 2-6).

A limitation has been added: '...some GP practices were asked to recruit by specific smoking status rather than the whole range of smoking status, potentially introducing bias to our sample.' (Page 22, lines 7-8).

(4) Reviewer's comments: p. 6, lines 17-20: How did participants consent? Did you attach a stamped return envelope?

Authors' response: Written consent was obtained in-person on the day of the interviews or focus groups, before the interview or focus group. The sentence has been amended to clarify this point: "Written consent and permission to audio-record were obtained on the day of the interviews and focus groups." (page 6, lines 32-33).

(5) Reviewer's comments: p. 7, lines 15-23: In the 'Participant recruitment and sampling' section, you state that high-risk patients were included in focus groups, but in the 'Study procedures' section this is not quite clear, as you only mention how healthcare professionals and community partners were contacted.

Authors' response: Thank you for highlighting this inadvertent error. The following text has been added to this section to clarify this point: "High risk members of the public, and healthcare professionals..." (Page 8, line 1).

(6) Reviewer's comments: p. 8, line 29: Can you specify which reading level test you used (e.g. Flesch Kincaid)?

Authors' response: Text has been added to page 9, lines 22-23 to clarify which reading test and formula was used: "Reading age was calculated using the Automated Readability Index (www.readabilityformulas.com)."

(7) Reviewer's comments: p. 10, line 6: Figure 1 and Figure 2 are tables. I suggest renaming this to avoid confusion.

Authors' response: Figure 1 and Figure 2 have been renamed as Tables (Table 2 and 4).

(8) Reviewer's comments: The results section would flow better and would be easier to follow if quotes were incorporated into the text rather than a separate table.

Authors' response: Although it would be our preference to incorporate the quotes into the text, due to limited word count, the quotes have been presented in Tables.

(9) Reviewer's comments: Throughout the results section, you should make it clearer that accounts of events/actions are self-reported by participants, rather than necessarily reflecting an objective reality. For example, "Symptoms that could indicate a chest infection were constantly monitored"; suggest changing to "Symptoms that could indicate a chest infection were reportedly constantly monitored". I suggest making this clearer throughout the results and discussion, as participants' accounts may be subject to various forms of bias (e.g. social desirability, recall...).

Authors' response: The results section (pages 11-18) has been edited to reflect the reviewer's comment; all changes have been highlighted in red text.

(10) Reviewer's comments: p. 14 line 14-17: "Healthcare professionals in Wales discussed new guidance that discourages health professionals from 'lecturing' patients, suggesting the patient reported experience may be based on previous healthcare interactions, and they consequently anticipate a lecture." Another possibility is that not all healthcare professionals are aware of or adhere to new guidance and continue to 'lecture' their patients. Based on self-reports alone, we cannot be sure which perception is more likely to reflect reality.

Authors' response: We agree with this point, and a sentence has been added to reflect this point: "Alternatively, healthcare professionals may be unaware of new guidance, or not adhere to new guidance and consequently continue to 'lecture' patients about smoking." (page 17, lines 11-12).

(11) Reviewer's comments: Discussion: p. 16, lines 9-11: "These recruitment and interview procedures meant we were able to explore actual and anticipated lung cancer symptom presentation in those who were symptomatic or asymptomatic." I would argue that this study does not allow an exploration of "actual" help-seeking, but rather accesses this through participants' self-reports. I would therefore suggest re-phrasing this, for example: "we were able to explore participants' accounts of previous and anticipated lung cancer symptom presentation"

Authors' response: Thank you for the suggestion. The sentence has been re-phrased: "These recruitment and interview procedures meant we were able to explore previous and anticipated lung cancer symptom presentation in those who were symptomatic or asymptomatic." (page 21, lines 32-34).

(12) Reviewer's comments: p. 17, line 18-19: "as a consequence of high lung cancer incidence in areas of deprivation." A reference is needed here to evidence the high incidence of lung cancer in deprived areas. Also, do you mean incidence or prevalence? I suggest prevalence may be the more appropriate term here.

Authors' response: A reference to the National Cancer Registration and Analysis Service report of cancer incidence by deprivation has been added to Page 21, line 18. The data report the association between deprivation and incidence; therefore, the term 'incidence' is most appropriate in this context.

(13) Reviewer's comments: p. 16, lines 4-17: The limitations of qualitative research should also briefly be mentioned here. While the methodology is appropriate to the research aim, causal links between factors (e.g. fear/denial and delays to help-seeking) cannot be established using this method. Furthermore, this methodology cannot establish whether certain processes are more common among deprived populations than non-deprived populations (e.g. normalisation of haemoptysis). Thus while the method appropriate it is worth mentioning the limitations.

Authors' response: The limitations of qualitative research have been discussed on page 22, lines 2-5: "However, our qualitative study was unable to establish causal links between barriers and help seeking, nor can we generalise or compare the findings to high socioeconomic groups; instead, we conducted an in-depth study to explore how best to engage high risk, highly deprived individuals in early lung cancer detection."

(14) Reviewer's comments: p. 16 line 31: You discuss that some participants felt they may not be eligible for treatment due to lifestyle. Corner et al ("Is late diagnosis of lung cancer inevitable? Interview study of patients' recollections of symptoms before diagnosis") report a similar finding, although only one participant in their study mentioned this. If more participants in your study discussed this, then you have a) found further evidence that this theme is of importance and b) it may be an indication that it is of particular importance among deprived groups (this would need to be confirmed using larger-scale, quantitative methods though). I suggest discussing this existing evidence here.

Authors' response: This section has been reworded, with the suggested reference included: page 20, lines 29-34: "Fear of being ineligible for treatment due to lifestyle has not been well described in studies with lung cancer patients or those at high risk[44,48]. In contrast, participants in the current study described feeling disintitled to medical services in the context of their lifestyle and circumstances. The underlying concept of health service Candidacy[47] may explain why participants felt unworthy of seeking medical help and is likely to be of particular importance in our highly deprived sample."

(15) Reviewer's comments: p. 17, lines 20-22: "However, we report normalisation and ignoring of haemoptysis, possibly due to a combination of high fear and fatalism of lung cancer, difficult life circumstances and low perceived health service Candidacy." A similar finding (i.e. that fear of cancer and fatalism may contribute to delays) was found by Tod et al ("Diagnostic delay in lung cancer: a qualitative study"). However, other studies (e.g. Corner et al, "Experience of health changes and reasons for delay in seeking care: A UK study of the months prior to the diagnosis of lung cancer") have found no obvious signs of fear or denial. Corner et al report that their participants seemed to minimise and normalise symptoms, but found no evidence that participants suspected lung cancer and delayed out of fear. It would be good to discuss your finding in light of this existing evidence. How is your study different, and what might account for these different findings?

Authors' response: We thank the reviewer for this comment; however, we think these references would be inappropriate to support this point. This section discusses normalisation of haemoptysis, and how fear and fatalism of lung cancer could potentially contribute to normalisation and ignoring of the symptom. The suggested references do not discuss normalisation of haemoptysis; instead, they discuss fear and fatalism more generally in relation to symptomatic presentation. The suggested papers were referenced elsewhere in the paper, when providing an overview of the evidence to date in the field, including how fear has the potential to deter help seeking.

(16) Reviewer's comments: p. 18, lines 10-13: "Finally, the current UK health system potentially encourages patients with a lung condition to focus on short term management of their condition. GP prescribing of antibiotics and the use of rescue packs may inadvertently reinforce patients to detect and act on symptoms of a chest infection." I think a little more explanation and some referencing is needed here. Are these known problems, and is there evidence in the literature that certain GP behaviours reinforce focus on short-term management?

Authors' response: NICE recommends the use of rescue medication for patients with COPD and regular exacerbations; a reference has been added to support this statement. The statement regarding how the current system of healthcare may reinforce focus on short term health was a reflection; we are not aware of literature to support this statement. In light of this comment, this section has been amended: "Finally, we suggest that the current UK health system may encourage patients with a lung condition to focus on short term management of their condition. GP prescribing of antibiotics and the use of rescue packs (prescribed antibiotics for storage at home in the event of an exacerbation) may inadvertently reinforce patients to detect and act on symptoms of a chest infection [50]." (page 22, lines 33-34 and page 23 lines 1-3).

(17) Reviewer's comments: p.17/18: The section on practice and policy implications could be improved by discussing your findings in light of existing interventions to encourage earlier diagnosis for lung cancer. There have been various community-based public awareness-raising interventions as well as GP-based interventions (e.g. Be Clear on Cancer, 3 week cough campaign...). How do your findings add to these? Do existing interventions address the issues identified by your study? How should existing interventions be adapted, according to your findings?

Authors' response: Thank you for your comment. We agree that this section could be strengthened by discussing our findings in relation to awareness interventions. The following text has been added: "Community based interventions have the potential to harness the relational aspects of help seeking, through interventions led by non-judgemental and welcoming facilitators. It is possible that previous mass media and social marketing lung cancer awareness interventions report low campaign reach to deprived groups [34,35] in part because they were not designed to motivate help seeking through intensive approaches to build trusting relationships and confidence. More research is required to understand how the relational aspects of help seeking could be operationalised in an intervention." (Page 22, lines 22-28).

Minor comments:

(18) Reviewer's comments: Abstract p. 2 Line 12: Word "in" missing (GP practices in socioeconomically deprived areas...)

Authors' response: The sentence has been amended: "Semi-structured qualitative interviews with 37 high-risk individuals, identified through seven GP practices in socioeconomically deprived areas of England, Scotland and Wales (most deprived 20%)." (Page 2, line 12).

(19) p. 2 Line 14-16: Sentence is incomplete (Four focus groups with high-risk individuals and local stakeholders (healthcare professionals and community partners) [were conducted] to explore preferences for an intervention to promote early lung cancer symptom presentation.)

Authors' response: The sentence has been amended: "Four focus groups with 18 high-risk individuals and 16 local stakeholders (healthcare professionals and community partners) were conducted to explore preferences for an intervention to promote early lung cancer symptom presentation." (page 2, line 14-17)

(20) p. 4, line 34: To ensure that our sample WAS at the highest risk of lung cancer

Authors' response: This bullet point has been amended: "A major strength of this study was the proactive and rigorous sampling procedures used, to ensure that our sample was at highest risk for lung cancer." (page 3, line 7).

(21) Reviewer's comments: p. 11, lines 7-9: "When symptoms were easy to detect, they were attributed to what was perceived as a treatable cause and represented an immediate health threat i.e. a chest infection due to lung condition, participants sought medical help quickly" There seems to be a word missing in the sentence (...lung condition, AND participants sought medical help quickly. ?)

Authors' response: The sentence reads as intended, therefore the sentence structure has been changed to remove ambiguity. The sentence now reads: "Participants reported seeking medical help quickly when symptoms were easy to detect, were attributed to what was perceived as a treatable cause and represented an immediate health threat i.e. a chest infection due to lung condition." (page 11, lines 32-34).

(22) Reviewer's comments: p. 12, line 22 – p. 13, line 2: "Therefore, the relationship with their healthcare professional was important when considering whether to present with lung symptoms; to feel understood by the healthcare professional, with their personal history taken into account in the context of health behaviour such as smoking." This sentence is very long and a bit awkward to read – suggest revising to improve the reading flow.

Authors' response: The sentence has been split, and re-worded to read: "Therefore, the reported relationship with their healthcare professional was important when considering whether to present with lung symptoms. Participants discussed the need to feel understood and not judged by their healthcare professional, with their personal history taken into account in the context of health behaviour such as smoking." (pages 13, lines 25-28).

(23) Reviewer's comments: p. 16, line 4: "A major strength of this study was the rigorous sampling procedures." Should be either "were the rigorous sampling procedures" or "was the rigorous sampling procedure".

Authors' response: The sentence now reads: "A major strength of this study was the rigorous sampling procedure." (Page 21, line 27).

Reviewer 2

It has particular value in that it points to multiple ways in which better engagement with and impact on, 'hard-to-reach' groups could be better effected. For example, the findings supports the view that a change in negative attitudes to the benefits and value of early diagnosis are likely to be as important in any campaign as raising awareness of symptoms; it adds weight to the finding of the ICBP about UK patients being inhibited from attending their GP for fear of wasting their time; it highlights the value of continuity of care by a single GP who, the patient knows and trusts; it emphasises the potential value of allowing people to seek health care advice in a more anonymous setting outside that of their normal General Practice.

I have no significant comments or suggested changes except for some relatively minor comments on parts of the introduction:

(24) Reviewer's comments: Introduction page 4, line 2 Latest 5 yr survival figure for England from ONS (Cancer survival in England: adult, stage at diagnosis and childhood – patients followed up to 2016) for patients diagnosed in 2011 was 14% for males and 17,5% for women.

Authors' response: Thank you for highlighting this error. The study was UK-wide; therefore, we have reported the latest UK-wide available lung cancer survival data (up to 2014). The % survival and reference has been updated: "Outcomes are among the poorest for all cancers, with only 13% of lung cancer patients surviving five or more years in the UK [2]." (page 4, lines 2-4).

(25) Reviewer's comments: Introduction page 4, lines 16-17 – mining is cited as an occupational cause of lung cancer. In a UK context this would be presumed to refer to coal mining and the evidence of an association between coal mining and lung cancer is very limited, with a much greater association being seen in quarrymen. The reference given by the authors for this is Malhotra (ref 12) and his study did not refer specifically to coal mining in the UK except where there is true silicosis. So, I think [coal] mining is a bad example to use and would suggest referring to asbestos exposure (using the same reference).

Authors' response: This sentence has been amended: "High prevalence of smoking, lung comorbid conditions and asbestos exposure, all of which are well documented risk factors for lung cancer, contribute to high lung cancer incidence and mortality in deprived communities[11, 12]." (page 4, lines 17-19).

(26) Reviewer's comments: Introduction page 4, line 23 – using the word 'emphysema' here is inappropriate since emphysema is one feature of COPD and there is no logic for referring to this specific, and essentially pathological, feature of COPD. So, I suggest simply changing the word emphysema to COPD.

Authors' response: This sentence has been amended: "...other comorbid conditions such as heart disease or COPD, thereby prolonging help-seeking[5,13,15-26]." (page 4, line 24).

Reviewer 3

I enjoyed reading this paper about early lung cancer detection in high-risk groups in UK's most deprived communities. This is a well-written paper with an original study design and insightful analysis, however it needs greater clarification regarding the methods of data collection and analysis and it would benefit from a refining of the discussion. Please see my detailed feedback and recommendations below.

(27) Reviewer's comments: Abstract: While the Abstract is overall well-written, it could benefit from some details for greater clarity: how many patients were recruited in the interviews vs. the focus groups? How many healthcare professionals and community partners? It needs to be made clear that the participants were cancer-free and that the symptom attribution task explored help-seeking intentions. The conclusions could be phrased more tentatively, e.g. 'community-based interventions might be a way to empower this population...'

Authors' response: Thank you for your comment. The abstract has been amended to reflect your comments. Please see page 2, lines 3-34.

(28) Reviewer's comments: Introduction: The Introduction draws on pertinent literature relating to lung cancer, delayed presentation and socio-economic inequalities, and fully justify the need to address what might motivate early symptomatic presentation among deprived groups. However, a justification is needed for why interventions are seen as the solution to delayed help-seeking and what these should consist of. The authors should include examples of interventions which have been developed to promote timely symptomatic presentation among high-risk individuals. The authors should clarify if there is an evidence gap around the patient factors that enable early presentation and include a stronger rationale for why a qualitative approach was undertaken and how it was expected to address this gap. The aim of the study could be phrased in ways that reflect the strengths of qualitative

approaches, e.g. 'This qualitative study aimed to understand the sense-making and motivations underpinning early presentation...' or 'what would motivate early presentation... and acceptance of interventions promoting early presentation.'

Authors' response: Thank you for your comment. We have added a sentence to clarify that there is a gap in evidence about early presentation in the very highest risk groups: "To date, research has mainly been conducted with lung cancer patients from a range of socioeconomic groups with varying levels of lung cancer risk, retrospectively exploring the barriers to symptom presentation. Evidence is lacking about how individuals who are at high risk, and without a diagnosis of lung cancer, attribute potential lung cancer symptoms and decide to seek medical help." (page 4, lines 27-31).

Due to limited word count, a brief paragraph has been added to justify why interventions are needed, and examples of previously developed interventions: "Strategies to prompt earlier help seeking for lung cancer symptoms are required. However, evidence is limited regarding optimal methods for promoting earlier detection through interventions targeted at high risk, highly deprived groups. Mass media[34] and community based social marketing[35] lung cancer campaigns report limited reach to the most deprived groups. A nurse-led primary care intervention for older adults with a long smoking history or recent cessation reported increased and sustained intentions to seek help with lung cancer symptoms[36]. However, the intervention was not targeted at highly deprived groups. Novel methods to support the high risk groups to engage in early lung cancer detection are required." (page 4, lines 33-34 and page 5, lines 1-6).

We have re-phrased the aim of the study to reflect the strengths of qualitative approaches: "The current study used a combination of interviews and focus groups to explore potential barriers to early lung cancer detection and strategies to encourage early help seeking with individuals who are the high risk for lung cancer. Qualitative interviews were used to gain an in-depth understanding of the processes and motivations involved in symptom attribution and medical help seeking for potential lung cancer symptoms in high risk, highly deprived individuals." (page 5, lines 8-12).

(29) Reviewer's comments: Method: Greater clarity is needed as to how many interviews and focus groups, respectively, were conducted, and with how many participants. Was cancer made explicit, and were any stimulus materials used in the focus groups? Under 'interviews', 7 high-risk individuals were recruited, yet 37 interviews were carried out. It would have been useful to include the topic guides for the interviews and the focus groups, respectively, as an appendix. The focus groups feel quite disjointed from the interviews, and there needs to be a clearer rationale for why the 37 interviews did not explore the content and format of potential interventions, yet separate focus groups did.

Authors' response: The number of interviews carried out has been reported on page 5, line 29: 'Thirty-seven interview participants...'. The number of participants recruited to the focus groups has been reported on Page 6, line 20: "Eighteen participants for the focus groups with members of the public..." And page 6, line 27: 'Local health service planning groups and health board staff facilitated recruitment of 12 participants for the healthcare professional and community partner (HPCP) focus groups.'

The numbers listed under the 'interviews' section of the methods refer to the number of GP practices involved in participant recruitment. This sentence has been re-worded for clarity: "Thirty-seven interview participants were recruited through seven primary care general medical practices (GP) in South Wales (Cwm Taf: three practices), England (Liverpool: one practice) and Scotland (Aberdeen: three practices)." (page 5, lines 29-31).

Focus group participants were explicitly informed that the study was about lung cancer. Text has been added to Page 8, lines 4-5 to clarify this point: "Focus group participants were explicitly informed that the study was about the development of an intervention about lung cancer."

We used focus groups to explore intervention preferences for a lung cancer intervention because the interview was framed around lung health, not lung cancer. The combination of interviews and focus groups allowed us to present key findings to the focus groups to consolidate findings and then explore intervention preferences. To clarify the synergy between the interviews and focus groups, the following text has been added: "To overcome methodological limitations associated with retrospective recall, we recruited participants with no previous diagnosis of lung cancer, and framed the interview around lung health, rather than lung cancer. Findings from the interviews were presented to focus groups participants in order to facilitate discussion about preferences and acceptability of interventions to engage high risk, highly deprived groups in early lung cancer detection." (page 5, lines 15-19).

In the analysis section, we discussed how overlap between data from the interviews and focus groups was handled (Page 9, lines 2-5).

The topic guides have been added to the appendix (Supplementary Files 2, 3 and 4).

(30) Reviewer's comments: The choice of Framework Analysis needs to be better justified and aligned with the qualitative approach and aims of the study, which seem rather exploratory in nature as presented in the Introduction. The use of the Common-Sense Model of Illness to conduct the interviews, on the other hand, suggests a rather deductive approach, hence the findings, using the Framework approach, could have been mapped onto the key dimensions of the model. The concept of data saturation, which originated in Grounded Theory, may not be appropriate within the Framework Analysis approach – the authors need to check this and rephrase as needed.

Authors' response: We welcome the extra words to describe more fully our reasons for choosing Framework Analysis and have added two sentences into the analysis section of the paper: 'Framework analysis is a well-respected and commonly used approach to qualitative data analysis. It was considered particularly suitable for this study due to its transparency and the team work involved[41]. Framework enabled the sharing of synthesised data charts among team members to facilitate participation in analysis and interpretation workshops.' (page 8, lines 30-34).

The Common-Sense Model was not used formally to conduct the interviews, rather it was part of the process used to inform the development of the topic guide. Text has been added to Page 7, lines 7-9 for clarity: "Interviews were conducted using a semi-structured topic guide to facilitate a discussion about illness perceptions and coping strategies; development was guided by the Common Sense Model[38] (Supplementary File 2)." We have added the study topic guide to the appendix (Supplementary File 2). The topic guide shows how the interviews were conducted and that the Common-Sense Model did not structure the interviews and that the interviews were exploratory in keeping with the study aims and the use of Framework Analysis.

The use of principles of data saturation in qualitative sampling is a widely used and well respected approach within qualitative sampling. Most qualitative quality appraisal checklists require consideration of saturation, as we have done using the COREQ checklist (see Supplementary File 1).

(31) Reviewer's comments: The analytic approach to the sorting task needs to be made more explicit and the authors need to report the results (or explain why these results are not included in the paper).

Authors' response: The purpose of the symptom task was to provide a concrete task to facilitate a discussion about symptom attribution and help seeking. The study was qualitative; therefore, we do report the ranks of each symptom on the task, although we considered the positioning of the symptoms when conducting the Framework Analysis. Text has been added to Page 9, lines 9-11 to clarify this point: "Although not formally incorporated into the analysis plan, the positioning of each symptom in the attribution task was considered during interpretive workshops."

(32) Reviewer's comments: Results and Discussion: The developed themes reflect well the quotes included, however the description of the findings needs to be more attuned to the qualitative paradigm. Phrases like "disparity between actual and anticipated medical help seeking was found for haemoptysis" and "some participants who had previously or were currently experiencing haemoptysis reported normalisation, leading to delays in medical help seeking or no help seeking" are rather unclear: had any participants sought help for symptoms indicative of lung cancer? How many had previously presented with lung cancer symptoms? How was 'delay' explored during the interview? It would have been interesting to know whether the high-risk participants spontaneously mentioned lung cancer and under what circumstances (given that the researchers avoided mentioning 'cancer' in the interviews).

Authors' response: Please see our response to reviewer 1, comment 9. We did not classify delay, or explore specific timescales during the interviews. Timescales were sometimes discussed spontaneously by participants when they disclosed how long (in time) they did, or would visit a healthcare professional for the symptoms during the symptom task. Some participants had experienced /were currently experiencing haemoptysis and did not seek help. The study was qualitative; therefore, assigning numbers of participants to our findings could be considered meaningless. We have added some additional text for clarity; however, our preference would be to omit numbers of participants from the findings: (Page 11, Lines 25-28): "However, some participants who had previously or were currently experiencing haemoptysis attributed the presence of blood to non-cancer causes such as their stomach ulcer or a previous flu jab. One participant ascribed their cough to lung cancer. Some of the participants with experience of haemoptysis did not seek medical help."

We have added the following sentence to clarify when participants mentioned lung cancer spontaneously: "For many participants, the topic of lung cancer arose spontaneously. Lung cancer was discussed in the context of perceived inevitability when reflecting on their general lung health and during completion of the symptom task when recalling friends/family with lung cancer." (page 12, lines 16-19)

(33) Reviewer's comments: Lines 20-21, p. 16: if the participants in this study were cancer-free, they could not have 'normalised their lung cancer symptoms' – perhaps rephrase as 'their symptoms indicative of lung cancer'.

Authors' response: This sentence has been updated: "Previous empirical studies report prolonged lung cancer symptom presentation due to misattribution [5,13,15-26,33,38,39] and in our study, we found evidence that participants normalised their symptoms indicative of lung cancer to smoking habit" (page 20, lines 19-20)

(34) Reviewer's comments: The assertion that this study was "the first to explore the influences on lung cancer symptom presentation and intervention preferences in high risk, highly deprived groups" is somehow not warranted given that intervention preferences were not explored during the 37 interviews with high-risk individuals.

Authors' response: Thank you, this sentence has been re-worded: "Our study was the first to explore the influences on lung cancer symptom presentation in high risk, highly deprived groups across three nations of the UK. Preferences for an intervention targeted at high-risk groups were ascertained through focus groups." (page 20, lines 2-4).

(35) Reviewer's comments: It feels like a missed opportunity that the useful concept of candidacy (Dixon-Woods et al., 2006) was not introduced earlier in the paper in relation to how people from low socioeconomic backgrounds seek medical help, or in the development of the themes. This would have made the themes more interpretative and would have added more theoretical layers to the analysis.

Authors' response: We agree with the reviewers comment; however, the concept of Candidacy was introduced during later analysis workshops when discussing consultation behaviour in high-risk individuals to facilitate the development of this theme. Therefore, we are unable to report the use of Candidacy in the development of interpretative themes, although we feel the concept is useful when interpreting our results in the Discussion.

Minor comments:

(36) Reviewer's comments: Line 12 in the Introduction: it would be more accurate to say that socioeconomic inequalities increase the likelihood of late presentation rather than the likelihood of a symptomatic individual to have lung cancer.

Authors' response: This point refers to the positive predictive values of symptoms, where a patient who presents to the GP with multiple symptoms, or has one or more of the listed risk factors is more likely to have cancer than someone with none of the listed risk factors. We have rephrased this point for clarity: "Multiple symptoms and risk factors for lung cancer including older age, smoking, the presence of a lung comorbidity and socioeconomic deprivation increase the likelihood that a patient presenting to their GP with symptoms indicative of lung cancer will receive a cancer diagnosis[7-9]." (Page 4, lines 11-13).

(37) Reviewer's comments: Lines 21-24 in the Introduction: the misattribution of vague symptoms may not be itself a cause of delayed help-seeking, but the fact that such symptoms are not considered legitimate reasons to seek medical attention.

Authors' response: Some text has been added to explain this point: "In the lead up to lung cancer diagnosis, vague symptoms may go unnoticed, not be considered a legitimate symptom to seek medical attention for, or be misattributed...." (page 4, lines 22-23).

(38) Reviewer's comments: Line 31 in the Introduction: given that the study was conducted with cancer-free participants, phrase the aims as exploring what might motivate early symptomatic presentation.

Authors' response: This has been clarified: "The current study used a combination of interviews and focus groups to explore potential barriers to early lung cancer detection and strategies to encourage early help seeking with individuals who are the high risk for lung cancer. Qualitative interviews were used to gain an in-depth understanding of the processes and motivations involved in symptom attribution and medical help seeking for potential lung cancer symptoms in high risk, highly deprived individuals." (Page 5, lines 8-12).

(39) Reviewer's comments: Perhaps replace the term 'fixate' with 'prioritize' or 'being preoccupied by', as fixation implies obsession.

Authors' response: The phrase 'fixate' has been amended to 'preoccupied' on Page 2, line 21: "Preoccupation with managing 'treatable' short-term conditions (chest infections)...", Page 17, lines 6-7: "In addition, the health professional group supported our findings that patients with lung conditions tend to be preoccupied by chest infections." Page 20, line 5: "high risk for lung cancer tend to be preoccupied by maintaining health in the short term".

(40) Reviewer's comments: Suggest moving the paragraph on the strengths and weaknesses of the study towards the end of the Discussion.

Authors' response: This paragraph has been moved to page 21, lines 27-34 and page 22, lines 1-11).

VERSION 2 – REVIEW

REVIEWER	Julia Mueller University of Manchester
REVIEW RETURNED	06-Nov-2018

GENERAL COMMENTS	I am happy with the way the authors have responded to and addressed my comments, and with the way the manuscript has been revised.
--

REVIEWER	Dr Afrodita Marcu University of Surrey, UK
REVIEW RETURNED	16-Nov-2018

GENERAL COMMENTS	I would like to commend the authors for addressing the reviewers' feedback and improving the manuscript which reads better now. Some minor comments below. Abstract Results: it should be made clearer where the findings come from, i.e. individual interviews or the focus groups. Conclusions: the broader contextual influences are perhaps on the appraisal of potential lung cancer symptoms rather than on their self-management. Introduction The paragraph describing the recruitment strategy (p.5, lines 15-19) belongs better in the Method section and should be moved there. (Also, the authors' response in the rebuttal letter for conducting focus groups separate from the interviews could be included in the Method itself as they provide a clear rationale for the study design.) Similarly, lines 20-22 p.5 belong better in the Discussion. Results Regarding the conceptualization and operationalization of delay in help-seeking, or of early presentation, it is true that numbers are not relevant here, however the authors could have provided a better explanation of how delay – or early presentation – was inferred from the participants' narratives. For example, what constituted a reasonable wait before considering contacting the GP for that symptom? The theme name 'denial and avoidance of longer term health' sounds rather strange and as if the participants were deliberately harming themselves. The authors need to make the theme name a bit more meaningful, e.g. 'Lack of consideration of long-term health', or 'Not thinking about own health in the future', if this is what it represents. However, some of the quotes included to illustrate this theme do not represent denial; on the contrary, the participants are aware of their own risk of getting lung cancer: e.g. "I've got [lung cancer] in my head, I'm probably going to get it because of the lifestyle I've had." (Male, 68, England, current smoker), or "[Lung cancer] is really, really on the forefront on the mind..." (Female, 81, Scotland, current smoker). These quotes do not seem representative of the theme.
---

	Discussion Similarly to my earlier comment about the theme on 'denial and avoidance of longer term health', greater clarity is needed about what the participants avoided. They avoided consideration of, or thinking about, lung cancer; they did not avoid lung cancer itself. P. 20, line 7, should read: avoidance of consideration of longer term health problems. The concept of candidacy (no need for capitalization) needs to be explained in text on p.20, as not all readers may be familiar with this concept (plus, the word has been used with different meanings in health research). Other minor comments: I wonder whether early lung cancer 'diagnosis' or 'presentation' rather than 'detection' would be more appropriate for the title and throughout the paper. I agree that 'incidence' is a more appropriate term than 'prevalence'. P.8, line 4: participants were invited to take part in focus groups, not 'in the focus group'. P. 8, line 20: members of the public were 'compensated with a voucher', not 'given'.
--	--

VERSION 2 – AUTHOR RESPONSE

Reviewer(s)' Comments to Author:

Reviewer: 1

I am happy with the way the authors have responded to and addressed my comments, and with the way the manuscript has been revised.

Authors' response: Thank you for re-reviewing our manuscript and for your positive response.

Reviewer: 3

I would like to commend the authors for addressing the reviewers' feedback and improving the manuscript which reads better now. Some minor comments below.

(1) Reviewer's comments: Abstract (Results): it should be made clearer where the findings come from, i.e. individual interviews or the focus groups.

Authors' response: Thank you for re-reviewing our manuscript and for your comments. Due to limited word count, we have added subheadings to the results section of the abstract to indicate the findings from the focus groups and interviews. "Interviews. Preoccupation with..." (Page 2, line 23) and "Focus groups. Participants recommended..." (Page 2, line 29).

(2) Reviewer's comments: Abstract (Conclusions): the broader contextual influences are perhaps on the appraisal of potential lung cancer symptoms rather than on their self-management.

Authors' response: This sentence has been amended. "This study was novel in engaging a high-risk population to gain an in-depth understanding of the broader contextual influences on lung cancer symptom presentation." (page 3, lines 2-3).

(3) Reviewer's comments: Introduction: The paragraph describing the recruitment strategy (p.5, lines 15-19) belongs better in the Method section and should be moved there. (Also, the authors' response in the rebuttal letter for conducting focus groups separate from the interviews could be included in the Method itself as they provide a clear rationale for the study design.) Similarly, lines 20-22 p.5 belong better in the Discussion.

Authors' response: The paragraph describing the recruitment strategy has been moved to page 6 (lines 2-4) and page 7 (lines 13-14). An adapted version of the response in the rebuttal letter has been added to page 5, lines 22-26: "We used a combination of interviews and focus groups because the interviews were framed around lung health (not lung cancer), whereas the focus groups were framed around preferences for a lung cancer intervention. In addition, key interview findings were presented in the focus groups for consolidation and to facilitate discussion about intervention preferences."

We thank the reviewer for their suggestion to move "To our knowledge, this was the first study to explore the influences on early lung cancer diagnosis and intervention preferences targeted at high risk groups living in the most deprived areas of the UK" from the Introduction to the Discussion section. However, we feel that this statement is best placed in the introduction.

(4) Reviewer's comments: Results: Regarding the conceptualization and operationalization of delay in help-seeking, or of early presentation, it is true that numbers are not relevant here, however the authors could have provided a better explanation of how delay – or early presentation – was inferred from the participants' narratives. For example, what constituted a reasonable wait before considering contacting the GP for that symptom?

Authors' response: When participants completed the symptom task, where possible, we explored actual or anticipated time to symptom presentation for each symptom. During our analysis workshops delayed or prompt help seeking was inferred based on the NICE guidelines for suspected lung cancer, guidance for patients through the NHS Choices website and the Aarhus statement. Any reference to 'delay' or 'early diagnosis' in the results section was removed from the previous transcript because we did not formally measure anticipated or actual delay.

(5) Reviewer's comments: Results: The theme name 'denial and avoidance of longer term health' sounds rather strange and as if the participants were deliberately harming themselves. The authors need to make the theme name a bit more meaningful, e.g. 'Lack of consideration of long-term health', or 'Not thinking about own health in the future', if this is what it represents. However, some of the quotes included to illustrate this theme do not represent denial; on the contrary, the participants are aware of their own risk of getting lung cancer: e.g. "I've got [lung cancer] in my head, I'm probably going to get it because of the lifestyle I've had." (Male, 68, England, current smoker), or "[Lung cancer] is really, really on the forefront on the mind..." (Female, 81, Scotland, current smoker). These quotes do not seem representative of the theme.

Authors' response: We thank the reviewer for their comment and for the suggested changes to the theme name. Participants described feeling that lung cancer was inevitable because of their lung condition or lifestyle, as reflected in the quotes above. However, participants were highly fearful and fatalistic about lung cancer; therefore, when combined with high perceived risk of developing lung cancer in the future, some described actual or anticipated denial or ignoring of symptoms and/or anticipated refusal of treatment. We agree that the findings appear contradictory; they perceive themselves to be candidate for lung cancer; however, because of high lung cancer fear and fatalism and complex social circumstances, their health is managed in the short term, leading to avoidance and ignoring of potential lung cancer symptoms and a preference to refuse treatment. The theme has been re-named to 'avoidance of acting on longer term health'. The text has been amended on page 2 (line 24), page 11 (line 3) and page 12 (line 20).

(6) Reviewer's comments: Discussion: Similarly to my earlier comment about the theme on 'denial and avoidance of longer term health', greater clarity is needed about what the participants avoided. They avoided consideration of, or thinking about, lung cancer; they did not avoid lung cancer itself. P. 20, line 7, should read: avoidance of consideration of longer term health problems.

Authors' response: Please see our response to the comment above. The text has been amended to reflect this comment: "Prioritising the daily management of their lung condition led to avoiding consideration of longer term health problems such as lung cancer, to gain a sense of control over health in the context of difficult personal circumstances." (page 20, line 7).

(7) Reviewer's comments: Discussion: The concept of candidacy (no need for capitalization) needs to be explained in text on p.20, as not all readers may be familiar with this concept (plus, the word has been used with different meanings in health research).

Authors' response: This sentence has been amended: "The underlying concept of health service candidacy (perceived eligibility for healthcare)[47] may explain why participants felt unworthy of seeking medical help and is likely to be of particular importance in our highly deprived sample." (Page 20, line 33)

(8) Reviewer's comments: Other minor comments: I wonder whether early lung cancer 'diagnosis' or 'presentation' rather than 'detection' would be more appropriate for the title and throughout the paper.

Authors' response: 'detection' has been amended to either 'diagnosis' or 'presentation' throughout the manuscript; all changes have been highlighted in red.

(9) Reviewer's comments: I agree that 'incidence' is a more appropriate term than 'prevalence'.

Authors' response: Thank you.

(10) Reviewer's comments: P.8, line 4: participants were invited to take part in focus groups, not 'in the focus group'.

Authors' response: This sentence has been amended: "...lung conditions were sent information about the study and invited to take part in focus groups." (page 8, line 9)

(11) Reviewer's comments: P. 8, line 20: members of the public were 'compensated with a voucher', not 'given'.

Authors' response: This sentence has been amended: "Members of the public who took part in the interviews or focus groups were compensated with a £10 shopping." (Page 8, line 25).

VERSION 3 - REVIEW

REVIEWER	Dr Afrodita Marcu University of Surrey, UK
REVIEW RETURNED	25-Jan-2019

GENERAL COMMENTS	I am satisfied with the way the authors have responded to the reviewers' feedback. I recommend this manuscript for publication.
---